# Assessment of High Performance Self-Consolidating Concrete through an Experimental and Analytical Multi-Parameter Approach

**DOI:** 10.3390/ma14040985

**Published:** 2021-02-19

**Authors:** Ghafur H. Ahmed, Hawreen Ahmed, Babar Ali, Rayed Alyousef

**Affiliations:** 1Department of Highway and Bridge Engineering, Technical Engineering College, Erbil Polytechnic University, Erbil 44001, Iraq; ghafur.ahmed@epu.edu.iq; 2Scientific Research and Development Center, Nawroz University, Duhok 42001, Iraq; 3CERIS, Civil Engineering, Architecture and Georresources Department, Instituto Superior Técnico, Technical University of Lisbon, Av. Rovisco Pais, 1049-001 Lisbon, Portugal; 4Department of Civil Engineering, COMSATS University Islamabad, Sahiwal Campus, Sahiwal 57000, Pakistan; babar.ali@scetwah.edu.pk; 5Department of Civil Engineering, College of Engineering, Prince Sattam Bin Abdulaziz University, Alkharj 16273, Saudi Arabia; r.alyousef@psau.edu.sa

**Keywords:** high performance concrete (HPC), self-consolidating concrete (SCC), flowability, durability, freeze-thaw cycle, fire resistance

## Abstract

High-performance self-consolidating concrete is one of the most promising developments in the construction industry. Nowadays, concrete designers and ready-mix companies are seeking optimum concrete in terms of environmental impact, cost, mechanical performance, as well as fresh-state properties. This can be achieved by considering the mentioned parameters simultaneously; typically, by integrating conventional concrete systems with different types of high-performance waste mineral admixtures (i.e., micro-silica and fly ash) and ultra-high range plasticizers. In this study, fresh-state properties (slump, flow, restricted flow), hardened-state properties (density, water absorption by immersion, compressive strength, splitting tensile strength, flexural strength, stress-strain relationship, modulus of elasticity, oven heating test, fire-resistance, and freeze-thaw cycles), and cost of high-performance self-consolidating concrete (HPSCC) prepared with waste mineral admixtures, were examined and compared with three different reference mixes, including normal strength-vibrated concrete (NSVC), high-strength self-compacted concrete (HSSCC), and high-performance highly-viscous concrete (HPVC). Then, a multi parameter analytical approach was considered to identify the optimum concrete mix in terms of cost, workability, strength, and durability.

## 1. Introduction

Self-consolidating concrete (SCC), also referred to as self-compacted concrete, is an innovative construction material with favorable rheological behavior that does not require vibration for placing and compaction. It can flow under its weight, filling in formworks, and achieving full compaction, even in the presence of complex-shaped concrete members with highly congested reinforcement [1,2,3,4]. Based on these properties, SCC may contribute to a significant improvement of the quality of concrete structures and opens up new fields for the application of concrete. The designation “self-compacting” is based on the fresh concrete properties of this material, which covers the mixture’s degree of homogeneity, deformability, and viscosity. The yield point defines the force required to make the concrete flow. The speed of flow of SCC is associated with its plastic viscosity which describes the resistance of SCC to flow under external stresses [5,6,7]. SCC has become a preferred option for many projects that should satisfy strict fresh stage properties and quality assurance. To ensure stable and robust fresh stage properties, typically, a significant amount of fine materials has been incorporated into the mixture. In relative to traditional concrete, different durability characteristics can be expected for SCC because it can be produced with various mix compositions and the absence of vibration [8,9,10]. Due to the relatively short history of SCC in practical applications, there is a significant lack of information about long term performance in real structures. Such a concrete should have a relatively low yield value to ensure high flowability, a moderate viscosity to avoid segregation and bleeding, and must maintain its homogeneity during transportation, placing, and curing [11,12,13].

High-performance concrete (HPC) is engineered to meet specific needs of a project, including mechanical, durability, or constructability properties. The demand for HPC has been continuously increasing due to its superior mechanical and durability properties [14,15]. When considering the cost of concrete production, HPC is even better than ultra-high performance concrete (UHPC), since heat-curing restricts the applications of the latter and makes it mainly suitable for precast elements, not for ready-mix concrete [16,17]. The development of HPC started in the 1980s, and thereafter the global demand for its consumption has significantly increased over the recent years. HPC can be designed to have high workability and mechanical properties as well as improved durability [18,19]. It has been primarily used in bridges and tall buildings. In general, durability is the most important parameter to increase the service life of any concrete structure [20,21,22]. Most commonly durability of concrete is affected by sulfate or chloride attack, carbonation, high temperature, and freezing and thawing damage [23,24]. Scanning electron microscopic studies [25,26] show that the pore structure in powder type SCC, including the total pore volume, pore size distribution, and critical pore diameter, is very similar to HPC. Over the past decades, advancements in concrete technology has led to the development of a new generation of concrete (e.g., HPSCC) with significantly better properties in terms of strength, durability features, and rheology of fresh concrete mixtures. In comparison to ordinary concretes, the designing process of HPSCC mix is determined by the increased cement content, superplasticizers, and an additive of reactive materials, i.e., silica fume. HPSCC is thus characterized by its ability to fill a form with congested steel rebars and self-leveling without mechanical compaction and it yields exceptionally high strength and durability [27,28].

Abundant research can be found in the literature on the properties of SCC. Most of the previous works have tested the fresh SCC mixes for common workability tests in order to prove self-consolidation of the concrete. The investigated properties were flowability, deformability and passing-ability, through slump-cone flow, J-Ring, V-funnel, and L-box tests [11,28,29]. The rheological properties of SCC such as yield stress and plastic viscosity [30,31] have also been investigated. Some researchers focused on the mix design and mix proportions [1,8,11]. The influence of mineral admixtures (i.e., silica fume, fly ash, metakaolin, ground granulated blast furnace slag, ladle slag) [32,33,34,35] and chemical admixtures (i.e., superplasticizers and viscosity modifying admixtures) [3,9,12] have also been studied on the performance of SCC. Some studies investigated the hydration rate and microstructure of SCC [1,13,17,36]. Researchers have also studied the properties of SCC an HPC with the addition of glass fibers, steel fibers and carbon nanotubes [6,18,37,38,39,40]. The stability tests results of SCC, i.e., shrinkage, cracking resistance, and creep are also available in literature [7,15,41,42].

A study reported that the elastic modulus, creep and shrinkage of SCC did not differ significantly from the corresponding properties of normal strength concrete (NSC) [43]. Some of the durability tests, including chloride penetration, water permeability and absorption, gas permeability, carbonation, electrical resistivity, sulfate attack, acid attack, frost resistance, and scaling, have been investigated [17,19,23,44] and more especially the fire resistance, cooling methods, weight loss, and residual mechanical properties of SCC [5,45,46,47]. Only few studies were found in the literature that investigated HPSCC [2,9,48,49] and its optimization [50,51,52]; these studies had focused on mechanical properties with either porosity, workability, water penetration, rheological properties, exposure to elevated temperature, or one durability test; but frost or scaling resistance of SCC have rarely been investigated in the literature.

Regarding the novelty of this work, it can be clearly seen in the literature that the HPSCC has been investigated in the past two decades, but its practical application is still limited. This is due to the fact that its consolidated technical performance of HPSCC (e.g., mechanical strength, durability, and cost) is not fully understood, and there are often insufficient statements concerning its exact overall behavior. Existing research provides information only about the improvements in the properties of HPSCC mixes through the variation in the composition or addition of materials, but it does not inform what will happen to other parameters such as its consolidated economic and engineering performance. In the view of this understanding, this research was designed to present a comprehensive study regarding HPSCC’s overall properties and comparing with three common reference concrete types, i.e., normal strength-vibrated concrete (NSVC), high-strength self-compacted concrete (HSSCC), and high-performance highly-viscous concrete (HPVC). The individual comparisons are based on the strength, workability and durability through 14 different types of tests. A new analytical approach has been proposed for a multi-parameter comparison between different types of concrete.

## 2. Material and Methods

### 2.1. Material Properties

The materials used for the concrete mixes were cement, silica fume, fly ash, fine and coarse aggregates, water, and superplasticizer. The cement was ordinary Portland cement type CEM-I 42.5R (, the micro-silica was MS90, which consisted of very fine SiO_2_ particles (up to 93.1%). The fly ash was type F, primarily consisting of silica, alumina, iron, and calcium oxides. The chemical and the physical properties of binders are shown in Table 1. The fine aggregate was normal fluvial sand, comprising the average passing percentages shown in Table 2. Fluvial gravel with a nominal maximum particle size of 12.5 mm was used in concrete mixes, and the average grading of 3 samples is shown in Table 2. High-performance superplasticizer concrete admixture Sika Viscocrete–5930 was used for obtaining workable or flowable mix made with a low water to cement ratio. The product was a third-generation superplasticizer with a density of 1.095 kg/L. Regarding the manufactures, cement, aggregates, microsilica, fly ash and superplasticizer were provided by Mass-Kurdistan company (Erbil, Iraq), Kalak quarry Hawler company (Erbil, Iraq), Jordan DCP company (Amman, Jordan), Jordan DCP company (Amman, Jordan), and Sika company (Istanbul, Turkey), respectively.

### 2.2. Mix Types and Mix Proportions

Design and selection of the concrete components is the most important step, which subsequently indicates the class and properties of the concrete. The intended concrete class was HPSCC, while three additional reference mixes were selected from 16 trial mixes. The reference mixes were HSSCC, HPVC, and NSVC. The considered four main optimization principles for better concrete production and mix design were workability, strength, cost, and durability. Table 3 can explain that 3 mixes were of the same proportions between cement, sand, and gravel, while NSVC is a conventional normal strength mix. The parameter that changed the HPSCC to self-consolidating concrete was the increased ratio of water, when compared to HPVC, since the binder-to-aggregate ratio was 0.24 for both mixes. Furthermore, the only difference that made HPSCC as high-performance concrete is the admixture type, when compared to HSSCC, as both mixes had the water to binder ratio w/b of 0.35.

### 2.3. Testing Fresh Concrete Properties

SCC is characterized by special fresh concrete properties. Many new tests have been developed to measure the SCC’s flowability, viscosity, filling ability, passing ability, resistance to segregation, self-leveling, and stability of the mixture. In this project, the conventional slump test, slump flow test, and J-Ring test were performed. The slump test is acceptable to determine the workability of non-flowable concretes having a slump of 15–230 mm when the cone is raised. When concrete is non-plastic or it is not adequately cohesive, the slump test is no more reasonable. The slump test was performed according to ASTM C143 for NSVC and HPVC mixes (see Figure 1).

HPVC had low water to binder ratio and more superplasticizer amount, therefore, the mix was very sticky, and needed additional effort for mixing, pouring, and casting. The slump flow test was performed for HPSCC and HSSCC, according to ASTM C1611 [53], to assess the flow rate in the absence of obstructions. During testing the accurate T_500_ (the time required for the slump flow patty to reach a 500 mm diameter) was recorded and when the concrete flow is stopped, the diameter of the spread at right angles is then measured and the mean is the slump flow (Figure 1). The restricted flow test was also performed according to ASTM C1621 for SCC classes. The J-Ring test represents the reinforcement inside the molds that restricts the flow of the concrete.

### 2.4. Testing Physical Properties of Hardened Concrete

Hardened density and absorption tests were performed for the four concrete mixes. The density of concrete was measured for different shapes and sizes and at different ages, in which the dimensions were measured to the accuracy of 0.01 mm, and the weights to 1 g. In the water absorption test, the concrete cubes were oven-dried at 60 °C for 48 h and the weights were recorded as oven-dry weights. After the cubes were submerged in water for 48 h, the surfaces were dried to represent saturated surface dry concrete.

### 2.5. Testing of Mechanical Properties

Strength tests are the most common for evaluation of different concrete classes; most of them were related to compressive strength by international standards. It is necessary to test as many as possible mechanical properties for special concrete classes, like HPC and SCC. To study the influence of shape and size of the specimens on compressive strength of different strength classes, 100 mm cubes and Ø100 mm cylinders were tested (Figure 2a).

The age of concrete was also considered, and the tests were performed at 1, 3, 7, 28, 56, 90, and 180 days. Splitting tensile strength was carried out on Ø100 mm cylinders, in which three cylinders were tested for each mix (Figure 2b). Another most common test for evaluating concrete’s tensile strength is the modulus of rupture. For this test, three prisms of 75 mm × 75 mm × 350 mm were prepared for each of the mixes and tested with 300 mm clear-span and third-point loading (Figure 2c). The compressive stress-strain relationship of concrete is the most basic constitutive relationship and is necessary for the understanding of structural response of concrete. The compressive stress-strain relationship was tested using Ø150 mm cylinders, that two cylinders for each of the mixes were tested (Figure 2d). Modulus of elasticity was calculated from the compressive stress-strain relationships.

### 2.6. Durability Tests of the Concrete Mixes

Heat resistance, direct exposure to the fire, freezing and thawing resistance, and scaling resistance were the tests carried out to assess the durability of the concrete mixes in extreme environments. The resistance of concrete to high temperature is one of the main characteristics of HPC mixes. The age of the concrete cubes of each mix at the time of testing was 36 days, and the maximum temperature of the oven shown in Figure 3 was 1200 °C. During exposure to high temperatures, the degree of strength-loss is dependent on the maximum temperature reached, heating/cooling rate, and the exposure duration. The heating rate was 200 °C/h up to 600 °C, 50 °C/h until 700 °C and whereas, the cooling rate was 25 °C/h. The specimens remained for 7.6 h at a temperature of +600 °C, and 2 h in +700 °C. 

A fire-attack is mostly considered as an accidental action, instead of a degradation process. For understanding the differences between direct fire resistance and oven heating, additional sets of cubes were subjected to direct fire (Figure 3). The test was performed for NSVC and HPSCC, and the average heating rate was 500 °C/0.5 h while the cooling rate was 95 °C/h. The specimens were exposed to direct fire for 1.70 h at a temperature of +400 °C and 0.75 h in +500 °C, with the maximum temperature reached, was 520 °C. The fire-temperature was regularly measured by a laser thermometer.

In this study, the freeze-thaw test was performed following the same procedure and temperature limitations in ASTM C666 [54], but only for 50 cycles, using 100 mm cubes, as shown in Figure 4a,b. The cubes were submerged in NaCl solution with a concentration of 40 g/L and then tested for loss in weight and strength at 225 days’ age so that the possibility of interference of chemical reactions in the microstructure of concrete can be eliminated. The scaling test is used to determine the scaling-resistance of a horizontal concrete-surface exposed to 50 freeze-thaw cycles in the presence of de-icing chemicals. It is intended to evaluate the concrete’s surface resistance qualitatively by visual examination as per ASTM C672 [55]. The prepared specimens for the tests were shown in Figure 4c,d of which, an aluminum frame was fixed to concrete specimens by a highly adhesive epoxy. Pans had an inside square dimension of 220 mm, and 25 mm was provided as a dike for the 6 mm depth of the solution.

### 2.7. Economic Assessment of the Concrete Mixes

Apart from the technical performance parameters, the cost is also an important factor to optimize the concrete mixes. In this study, the cost of concrete mixes was calculated without VAT (taxes). The data for economic assessment considerably vary between regions. This is because local conditions highly affect the cost of labor, and the market costs for recovered materials, as well as the transportation scenarios. In this study, the most probable case scenario for the city center (Erbil, capital of Kurdistan region in Iraq) was considered to estimate the cost of the concrete mixes. The distance between the concrete plant and the raw materials, namely cement and aggregates was 184 km and 110 km, respectively. Besides, the other raw materials are imported from Turkey.

## 3. Results and Discussions

### 3.1. Slump Test

The settlement time of NSVC was different from that of HPVC since the latter had contained the superplasticizer; it outspread at a slower rate until 8 s after the lifting of the cone, while NSVC was stable within 3 s. The slump test results were shown in Table 4. Both results were considered acceptable for good workability during casting of concrete, while here the weak point of the slump test can appear when the HPVC was behaving acceptable for the slump test, but the mixture was very stiff that could not perfectly fill the mold without extra vibration. 

### 3.2. Flow Test

The slump flow test results were shown in Table 4. The flowing of both concrete types was such that neither bleeding nor segregation had occurred. The flow diameter of micro silica concrete (HPSCC) was 810 mm with T500 of 2.62 s, while the flow diameter of fly ash concrete (HSSCC) was 750 mm, which was lower by 8%, but the T500 was 3.8 s and was higher by 44%. The results also proved that the micro-silica helped in achieving a better flowability (in addition to the higher strength) because the fly ash particles were relatively larger compared to microparticles of the silica. Research showed that the additional grinding of fly ash did not cause workability loss of SCC and the plastic viscosity has increased [31,56].

The flow of 740–900 mm is used as a conformity limit for highly congested structures, and the conformity criteria of 2–5 s is used for the T500 when the improvement of segregation-resistance is necessary. T500 of less than 2 s is applied for very congested structures, better surface finishing, and risk of bleeding or segregation. From a practical point of view, increasing the initial flow head can also increase the flow energy necessary to transport coarse aggregate [1,56]. The ability of SCC mixtures to resist segregation was determined based on the assigned segregation index (SI). If there is no obvious accumulation of coarse aggregate particles and no free water flowing around the concrete’s perimeter, the mixture is assumed to have full segregation resistance (SI = 0). If the mixture exhibited an apparent accumulation of coarse aggregate or a small amount of water flowing, the mixture is unlikely to segregate (SI = 1). In case of obvious accumulation of coarse aggregate or free water, the SCC is likely to segregate (SI = 2). Finally, a large amount of accumulated coarse aggregate or a large amount of free water flowing indicates that the concrete will segregate, and the mixture must be rejected [25]. 

### 3.3. Restricted Flow Test

Results of the J-Ring test were shown in Table 4, noting that the restricted flow can decrease the flow diameter and T500, while the rate of restriction in the micro-silica concrete (HPSCC) was more than that of the fly ash containing concrete (HSSCC) compared to unrestricted flow. The difference in height inside and outside the ring was clearly showing a better flowability of the HPSCC, since the thickness of the concrete along the diameter of flow was almost homogeneous, neither bleeding nor segregation were observed. Restriction of concrete flow was causing the reduction in flow diameter by 6.2% and 1.3%, respectively for HPSCC and HSSCC, but the HSSCC mix had exhibited a little variation inside and outside the ring, not reaching the limit of segregation. The difference between the T500 values measured using the J-Ring test and the slump flow test should not be more than 2–4 s according to ASTM C1621 [57].

### 3.4. Density of the Hardened Concrete

Concrete density is an important property that is used in the design of concrete-structures through calculating the self-weight of the members. Table 5 shows the concrete density for the four concrete mixes considering the age of concrete. It can be observed that concrete densities were decreased with time. This phenomenon can be justified by continuous chemical reactions inside the concrete structure. The ratio of weight loss was between 0.5 to 2.5% when comparing 28-day densities with that of 6 months. It can also be noted that the two SCC mixes had less standard deviation (SD) than that of vibrated concrete mixes. When considering the shape and thickness of the concrete, neither systematic relation nor clear differences could be found between tile shaped specimens and the cubes, when comparing the 28-day densities in Table 5.

### 3.5. Water Absorption by Immersion

Results of the water absorption test are shown in Table 5, and each value in the table was representing the average of 4 tests. The results show that the water absorption of the NSVC was the largest due to the high void ratio, larger particle sizes, and less binder content compared to other mixes. Very low water absorption was observed in the HPVC mix due to the high binder content and improved packing of the particles as a result. Water absorption of the NSVC was 11% higher than that of the HSSCC; it was 2.4 and 4.1 times higher than that of HPSCC and HPVC, respectively. Similar results of about 2% water absorption were obtained in the previous study [28], while for higher fly ash replacements of 70 and 90%, the absorption was increased to 3.5 and 4.7%, respectively. Research showed that for an HPC with a water–binder ratio of 0.40 at 28 days, the water permeability was about 9 times higher than an HPC with a water–binder ratio of 0.23 [17].

### 3.6. Compressive Strength

The compressive strength results shown in Table 6 are average values of three 100 mm cubes. The rate of gaining strength is different between concrete types; the ratio of gaining strength at earlier ages (1 day) was 8% for NSVC, but it was 21% and 25% respectively for HPSCC and HPVC. At 28 days NSVC gained two-thirds of its 90 days’ strength while the ratio was 72%, 79%, and 85% respectively for HSSCC, HPSCC, and HPVC type mixes.

No considerable differences were observed at 180-days. The water/binder ratio has a great influence on the compressive strength of SCC and VC, whereas, the subject is still controversial and the authors got different conclusions. Some studies on the mechanical behavior of SCC showed that for the same w/b ratio, SCC has generally lower mechanical strengths than traditional vibrated concrete [2]. However, other studies stated: Compared with the majority of the published test results the tendency becomes obvious that at the same w/c ratio, higher compressive strengths were reached for SCC [12]. Three cylinders with Ø100 mm had also been tested for each of the mixes, to study the influence of the shape of the specimen on compressive strength of concrete mixes. Results showed that the higher compressive strength mixes were less affected by specimen shape since the cube to cylinder factor for NSVC mix was 0.79, but it was 0.90 and 0.96 for HPSCC and HPVC mixes, respectively. The shape of specimens and loading direction during tests, were the two factors that controlling (fcy/fcu) ratio. Other authors have reported that the compressive strength ratio of cylinders to cubes is 0.80–0.85 for VC, but it is 0.90–1.00 for SCC [7]. Silica fume is the most commonly used admixture for the production of HPSCC. It has been reported that adding 10% silica fume to the mixtures can increase the compressive strength by 30–100%, 6–57%, or 5–24%, by different authors [15,29,43]. The ratio of the fly ash used for HSSCC was negatively affecting the compressive strength of the mix, since, it is determined that the optimum fly ash content is 25–35%. In essence, fly ashes with 10% do have positive influence on overall quality of SCC, which increases the workability, frost durability and an acceptable level of strength. Further increase in FA% led to reducing of the CaO content, which led to a lower level of hydration [10,25,33].

### 3.7. Splitting Tensile Strength

The average results of three 100 mm × 200 mm cylinders that tested for splitting tensile strength are shown in Table 7. The tensile strength of HPSCC was 1.53 times that for NSVC, while HSSCC had a tensile strength of only 14% higher than NSVC, remembering that its compressive strength was 67% higher. When comparing the ratio of tensile strength divided by the square root of compressive strength, NSVC had a value of 0.62, but a higher value of 0.65 was recorded for HPSCC mix, 0.66 for HPVC, and again HSSCC was lower and it was only 0.55. Piekarczyk recorded a ratio of 0.61 for an NSC and 0.65 for an SCC [1]. Others stated that the relationship between tensile and compressive strength of SCC is similar to that of VC [2]. Research showed that the average direct tensile strength of the SCC was found to be 3.5 MPa, whilst the average splitting tensile strength was found to be 3.8 MPa, which is only 8.6% higher [58]. 

### 3.8. Flexural Strength

The average test results of three 75 mm × 75 mm × 350 mm prisms at age of 90 days were presented in Table 7. HPSCC developed a flexural strength 15% lower than that of HPVC, while it was about 50% higher than both HSSCC and NSVC. The ratio was 0.75 in the case of HPSCC; it was 0.68 for an NSVC and 0.78 for an HPVC. When comparing the results of splitting tensile strength and the flexural strength, it can be noted that the concrete prisms with the higher strength behaved better against tensile stresses than the splitting cylinders.

### 3.9. Stress-Strain Relationships 

The response of the concrete against stresses is different for the four mixes; when the strength of the concrete is higher, the strains were smaller for the same load level. Stress-strain relationships for the concrete mixes were shown in Figure 5. Two cylinders were tested up to 70–90% of the failure load for each of the mixes. The results are showing that the higher strength concrete mixes, especially for HPSCC and HPVC mixes were going in a straight path up to 80% of the load, whereas the case is not similar for NSVC, as it was starting deviation from linearity in the earlier stage of about 40% of the applied load. In 30 MPa stress level, and when comparing other mixes with HPSCC, the strain was higher by 26% and 100% for HSSCC and NSVC respectively, while the strain was less in HPVC by 12%.

### 3.10. Modulus of Elasticity

Determination of modulus of elasticity was based on the 40% of ultimate load and 0.000050 strain level. The obtained results were arranged in Table 7. When comparing the values in the table, it can be noted that the elastic modulus is not only the function of the compressive strength, since the composition of the mixtures plays a great role. The elastic moduli of the HPVC and HPSCC were close in value despite their different compressive strengths. The ratio of E/√*f*′c was 4.49 and 4.27 for HPSCC and NSVC mixes respectively. The recorded ratio by [6] was 4.20 and 4.33 for an SCC and NSC, respectively. As it is known, the modulus of elasticity depends on the proportion of Young’s modulus of the individual components and their ratio by volume; thus, the modulus of elasticity of concrete increases with a high content of aggregates of high rigidity, whereas it decreases with increasing hardened paste content, and increasing porosity [7]. On the other side, packing of the particles and optimization of the mix composition leads to a higher elastic modulus, even with fewer or no coarse particles as in the case of UHPC [16]. Research showed that the modulus of elasticity of SCC seems to be very similar to that of VC, with an important but similar scatter present on the results for both types of concrete [59]; other authors concluded that the reduction in the elastic modulus of SCC compared to VC is 5% for SCC with high compressive strength (100 MPa) and up to 40% for those with the lowest strength (20 MPa) [2]. Meanwhile, the modulus of elasticity of SCC specimens with SF is increased with SF content increase [25], while it sensitively decreases with an increase in the FA replacement ratio [33].

### 3.11. Oven Heating Test

Concrete is a composite material that derives properties from its multiphase and multi-scale ingredients. These ingredients are thermally inconsistent and during fire conditions, start to dissociate, leading to degradation in its strength and durability. Although, the behavior of HPSCC subjected to fire has not been extensively studied and thus remains largely unknown [60]. The heating of concrete may be advantageous or causing a reduction in strength. Research showed that concrete specimens exposed to 300 °C might have an increased compressive strength by 18–22%, especially in earlier ages [2,16]. This increase refers mainly to the acceleration of the hydration process at an increased temperature. The limit at which transfers the heating of specimens changes from a useful to destructive factor depends on many parameters; however, in general, temperatures higher than 400 °C are regarded as destructive. In this study, the temperature of the oven was 700 °C, to prevent the explosion of the specimens. Results the Table 8 show that the loss of compressive strength was 79%, 63%, 52%, and 49%, respectively for NSVC, HSSCC, HPSCC, and HPVC. Thus, when a NSC exposed to +700 °C for 2 h it can resist only one fifth of its designed load, but HPC can still resist half of the load. The results of residual compressive strength were within the range of the database presented graphically in [47] which includes limits of codes and researchers’ data. After initial heating to 400 °C in [5], the compressive strength decreased by 41–48% for an HSC containing 12.5% of silica fume. At 600 °C and 800 °C, the loss in strength was up to 44% and 79%, respectively. The strength loss at 400 °C was up to 18% whereas, at 600 °C and 800 °C, the strength loss was around 44% and 76%, respectively [19].

When concrete is exposed to a gradually increased temperature, the heat was transferred from the outer face of the concrete to its core, this process requiring less time for NSC and more time for HPC. The weak point of HSC classes is in that the heat was restricted by the dense microstructure, which leads to an explosion and spalling of concrete corners due to pore pressure, but the root cause of the failure was the cracking of concrete due to thermal tensile stresses, and the specimens with higher tensile strength can resist more pore pressure and spalling stresses. The relatively loose microstructure of NSC leads to absorption of heat to the concrete core and disintegrating its structure mainly due to the pore pressure build-up and the development of thermal stresses. Gravel particles can easily pull out and the burned paste is similar to dust, crushable with fingers, as shown in Figure 6, while in the case of HPC, the core of the cube is safer and the bond is still strong. The weight loss of 5–8% is recorded in this study, which is similar to that found by other authors. A mass loss of 4–6% was observed in [46] for 9 different mixes subjected to 1000 °C and last in the furnace for 90 min, and the mass loss of 2–9% has been reported in [19].

### 3.12. Fire Resistance Test

Subjecting of concrete specimens directly to the fire is different from oven heating, regarding the distribution of the heat around concrete faces. Fire test results on +500 °C for 45 min were shown in Table 8. The loss of compressive strength for NSVC was 34% and it was 22% for HPSCC. Reduction in the strength of the cubes under fire was less than one-half when compared to the heating of the cubes in the oven; however, the main reason for these smaller reductions was the lower level of heating. Generally, the temperature which makes an NSC have a poor strength is in the range of 600 °C. The strength degradation is primarily ascribed to decomposition of hydration products, such as, calcium silicate hydrates, calcium hydroxide, and carbonates. Le et al. reported that HPC loses up to 50% of its ambient temperature strength at 600 °C [46]. The pore pressure development in HPC samples is much faster than in SCC samples. The moisture content, the dense microstructure, and the tensile strength are the main influencing factors that determine the spalling of HPSCC. Research showed that the critical pore diameter of SCC is bigger than HPC; therefore, SCC will have larger damage once exposed to fire. When exposed to the fire of 200 °C for 18 min, the highest pore pressure at 10 mm depth of HPC was 2.52 MPa; while in SCC it was 1.27 MPa [46].

### 3.13. Freezing and Thawing Cycles

The test results of the freeze-thaw cycles were evaluated through the mass loss of concrete and the residual compressive strength, as presented in Table 9. HPCs showed negligible mass loss of 0.02%, while NSVC exhibited a drastic loss of 83%. HPSCC had lost 9.7% in the compressive strength, whereas, NSVC was almost damaged by losing 86% in compressive strength. HSSCC had lost 6% of its weight and 37% of its strength (the mass loss of this type of specimen was primarily in the top surface, which had less relative density).

The changes in concrete surface and the corresponding number of freeze-thaw cycles were shown in Figure 7. Deterioration processes typically begin when; aggressive fluids penetrate through capillary pore structure to the reaction sites where they trigger chemical or physical deterioration mechanisms [61]. When the test was running, in the first 10 cycles, the NSVC corners were subjected to the internal tensile stress. Later when the frozen salty water was causing volumetric internal pressure on the concrete surface, gravel particles started appearing and then got pulled out. If the pores are critically saturated, water will begin to flow to make room for the increased ice volume. The concrete will rupture if the hydraulic pressure exceeds its tensile strength. The cumulative effect of successive freeze-thaw cycles is the disruption of paste and aggregate eventually causing deterioration of the concrete. HPCs were resisting pore pressure due to their high tensile strength. In this test, loss in dimensions or lose of the concrete cover seems to be logically more acceptable when considering large structural members. The thickness loss was 25–30 mm in NSVC, 2–3 mm for HSSCC and the rest of the cubes almost had no thickness loss as in Figure 7. Similar deteriorations of NSVC and HSSCC concrete cubes have been observed in [30].

Water absorption is a key parameter in the investigation of the durability of concrete. Because of its low w/b ratio of 0.20–0.45, it is widely believed that HPSCC should be highly resistant to both scaling and physical breakup due to freezing and thawing. Research showed that non-air-entrained HPC with w/b 0.22–0.31 could be extremely resistant to freeze-thaw damage and it was suggested that air-entrainment and supplementary cementitious materials are not needed. Among six mixtures tested; only the silica fume concrete with w/b 0.22 was frost resistant [8]. The weight change is an indication of the deterioration of the concrete specimen. Weight change of 0.3–5.3% recorded in [35] and 2.0–56.5% is recorded by [30] for 13 SCC mixes.

### 3.14. Scaling Test

The concrete specimens were exposed to 50 cycles of freezing and thawing. NSVC lose weight of 372.6 kg/m^3^ and the aggregate particles appeared in early stages on the entire surface of the concrete; in HSSCC, initially, the first layer of concrete surface wore at earlier stages, but later, the degradation of the concrete surface was almost stopped or it was wearing very slowly so that the total weight loss after 50 cycles reached 27.2 kg/m^3^; both HPSCC and HPVC mixes were durable, showed no scaling, and a negligible loss of weight by having 0.387 and 0.286 kg/m^3^ respectively, as shown in Table 9. Mass loss of 0–0.5 kg/m^3^ after 50 cycles is recommended for HPC. Rating of specimens was performed as in ASTM C672 [55]. Scale rating of 0–1 after 50 cycles is recommended for HPC. The rating results for important checkpoints were shown in Table 10. The test is qualitative, and the rating was decided with a visual examination based on the surface of two specimens. For HSSCC, For HPCs, the top thin paste layer or was resisting the wearing and was not removed until the end of the test; only several small dark spots appeared as shown in Figure 8. Rating of the concrete surface can be evaluated by loss of concrete mass and visibility of gravel particles, whereas the interesting parameter in a practical point of view is the thickness of the deteriorated concrete, therefore it is better to determine the loss in thickness of the concrete which exposed to freezing and thawing cycles. Table 9 is also showing the concrete depth lost by the action of thermal stresses.

When comparing the scaling test results and the results of freezing and thawing test cubes, NSVC had lost 15.8% of its weight in the scaling test, but the loss was 82.6% in concrete cubes. This difference can be justified by the that the cubes were entirely submerged in water and attacked all sides, but the scaling pans were exposed to freeze-thaw cycles only at the top surface. Gagne et al. tested 27 mixes using silica fume with w/b of 0.23, 0.26, and 0.30, and a wide range of air–void systems. All specimens performed exceptionally well in salt scaling, confirming the durability of HPC. Also in [8] the weight loss of 0.1–4.5 kg/m^2^ is obtained after 40 cycles of scaling, for 3 concrete types with an air content of 2, 4, and 6% and w/c ratio of 0.25–0.50.

## 4. Hexagonal Model for Cost–Strength–Workability–Durability Relationship

An increased upfront cost must be expected for HPSCC, especially for better performance, enhanced quality control, along with better-quality formwork to withstand higher pressures. On the other hand, the incorporation of industry by-products increased productivity, reduced labor, and energy consumption, and fewer post-construction repairs can balance the final cost. The HPSCC is much more economical in certain applications on a basis of the original cost, and also in a point of view the durable upkeep, and more ecological than the usual concrete. Moreover, the lifespan of HPSCC is estimated at two or three times than on a usual concrete. The major obstacles that prevent a wider implementation of HPSCC in construction are its high cost (+25 to 50%), and the lack of knowledge of the properties. In Table 11 the major parameters (strength, workability, durability, and cost) are been considered simultaneously on a scale of 0–10. The strength and durability results were converted based on the highest value obtained among the four concrete types.

For comparing the workability results, SCCs were considered as perfectly workable, while the vibrated concretes can achieve 6 points for a 200 mm slump. Decreasing the slump has to decrease in the scale by one point for each additional 25 mm. The cost is negatively influencing the concrete selection; therefore, the lowest price was divided by the cost of other concrete types and multiplied by 10. The shown values of the concrete cost are based on the local prices for the preparation of 1 m^3^ of concrete. The average results of multi-parameter assessment scale (MPAS) are showing that the HPSCC is the best option (MPAS = 196.6) in selecting a mix among the tested classes (Figure 9).

## 5. Conclusions

The current study focuses on the assessment of HPSCC through an experimental program including 14 major performance tests that have mostly been used individually in the previous studies, and concluding improvement of some concrete properties, while some properties questionably remained if they were improved or worsened. A comparative and comprehensive study on the HPSCC can guarantee and answer that, what will happen to other properties. It was intended to apply the hexagonal model to some experimental studies in the literature, while the multi-parameter experiments (workability, cost, strength, and durability) can hardly or never be found. Therefore, it is of great interest to investigate such a type of comprehensive experiments. Based on the results and discussion, the following conclusions can be drawn:i.HPSCC exhibited excellent workability and flowability compared to NSVC and HPVC mixes that need vibration or compaction effort, hence, HPSCC is eco-friendly and beneficial for the environment by using sustainable materials like micro-silica and fly ash.ii.The density of the HPSCC was almost in the range of normal concretes, whereas the absorption of HPSCC was less than one half of both HSSCC and NSVC.iii.HPSCC was an early strength-gaining concrete by obtaining 20.9 MPa of compressive strength in one day, and 100.2 MPa at 90 days, which is higher than that of NSVC and HSSCC by 86.9% and 20.9%, respectively. The tensile strength of HPSCC from both splitting tensile strength and flexural resistance was much higher than the two mentioned mixes. Thus, due to its strength HPSCC can be useful in decreasing structural section sizes, and thereby the amount of concrete and cement used in construction projects.iv.HPSCC showed an elastic modulus of 42.7 GPa and had better resistance to the compressive strains and deformation, compared to those of NSVC and HSSCC. The relationship of the stress-strain curves was linear for HPSCC up to 80% of the ultimate load, while the curve for NSVC was starting deviation from linearity in an earlier stage of about 40% of the applied load.v.When exposed to fire flame or high temperature of 700 °C for 2 h, HPSCC behaved as a durable concrete, and had a residual strength of 48.2%, while the residual strength of NSVC and HSSCC were 21.4% and 37.3%, respectively. So, during fire accidents, HPSCC can survive more, and the probability of demolishing after the fire and new construction is much less.vi.Freeze-thaw 50 cycles were causing degradation in the compressive strength of HPSCC by only 9.7%, while the strength loss of NSVC and HSSCC was 85.6% and 36.8% respectively. In the scaling test, the average thickness losses for HPSCC were almost negligible, while the thickness loss of NSVC and HSSCC was 8 mm and 1 mm, respectively.vii.The proposed hexagonal comparison model can successfully predict the most beneficial concrete mixes, considering workability, strength, cost, and durability.

Even though this study considered the main parameters of concrete such as technical properties and cost, further studies are still required on this path since the optimization (final outputs) was made based on 1 m^3^ of concrete. Therefore, further studies on the mentioned parameters must be done by considering the real structural application such as beam and column.

## Figures and Tables

**Figure 1 materials-14-00985-f001:**
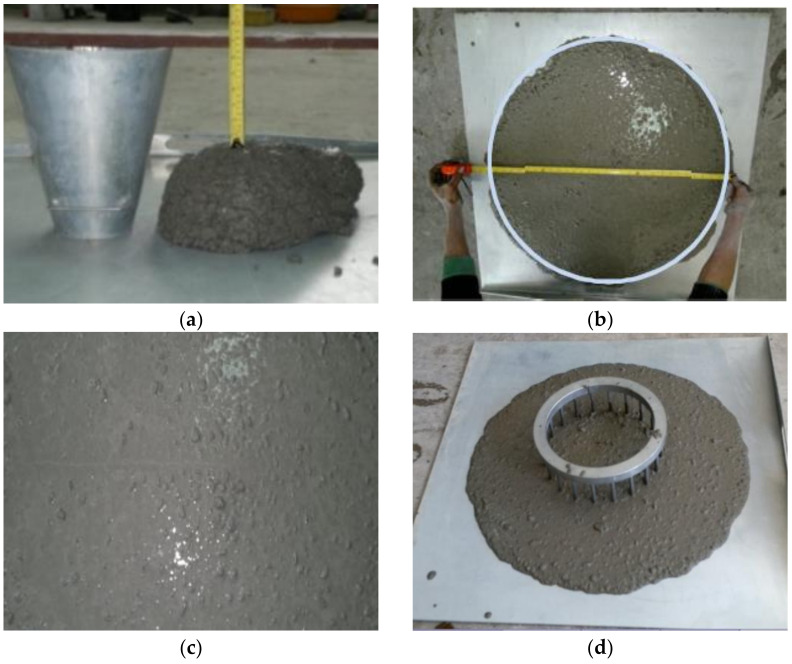
Fresh properties tests (**a**) slump test for vibrated concrete mixes, (**b**) slump flow test for self-consolidating concrete (SCC) mixes, (**c**) a SCC without segregation, and (**d**) restricted slump flow test.

**Figure 2 materials-14-00985-f002:**
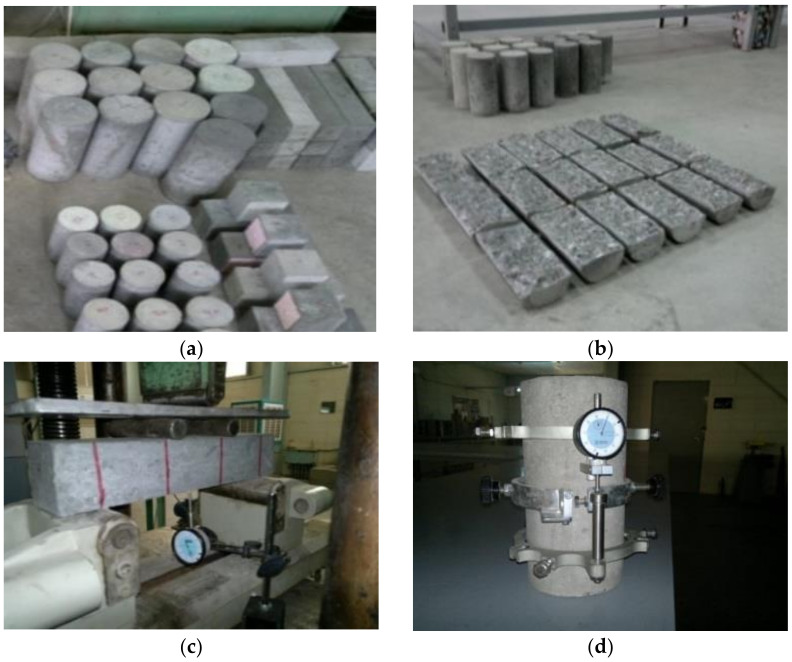
Mechanical properties tests (**a**) various size and shape specimens for compressive strength, (**b**) cylinders in splitting tensile strength test, (**c**) flexural strength test of concrete specimens, and (**d**) testing modulus of elasticity for Ø150 mm cylinders.

**Figure 3 materials-14-00985-f003:**
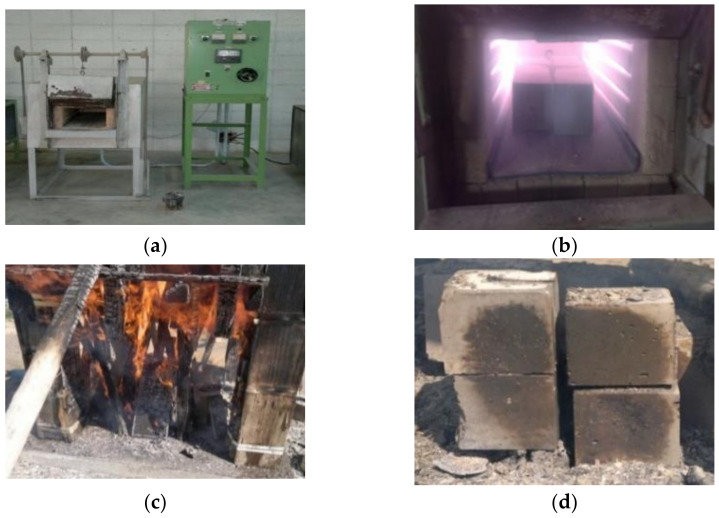
Heating of specimens. (**a**,**b**) Heat resistance test for concrete 100 mm cubes. (**c**,**d**) Exposure to direct fire flame test for concrete cubes.

**Figure 4 materials-14-00985-f004:**
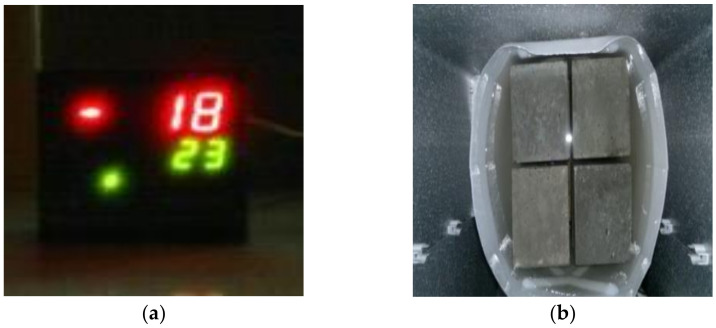
Freeze-thaw testing setup. (**a**,**b**) Freeze-thaw test for concrete cubes. (**c**,**d**) Scaling test for concrete specimens with aluminum frame.

**Figure 5 materials-14-00985-f005:**
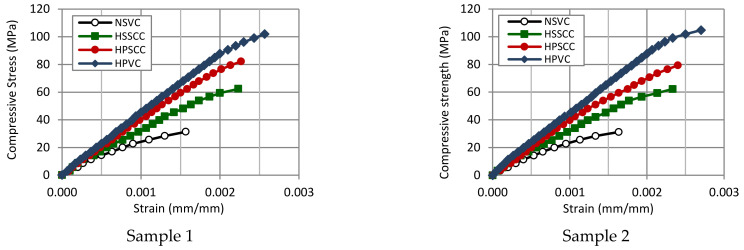
Stress-strain relationships for the four concrete mixes.

**Figure 6 materials-14-00985-f006:**
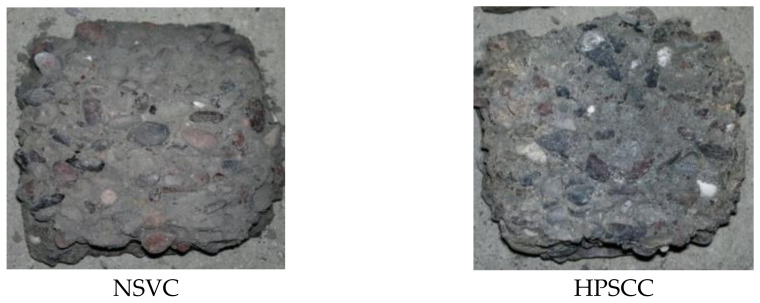
The cleaned core of broken cubes heated to 700 °C.

**Figure 7 materials-14-00985-f007:**
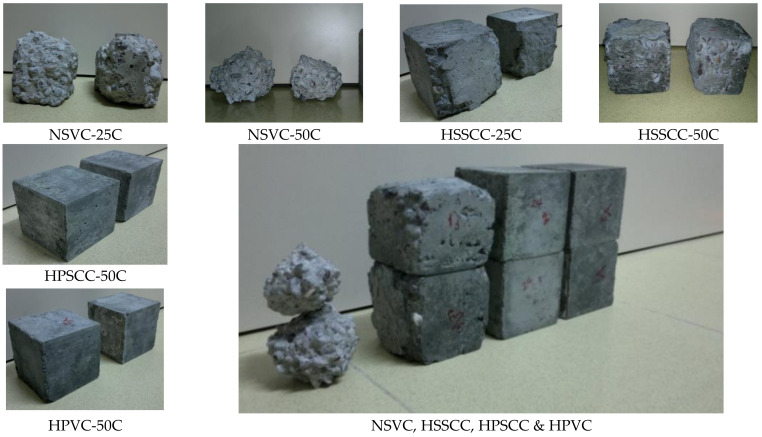
Freeze-thaw specimens after 25 and 50 cycles for NSVC and HSSCC and 50 cycles for HPSCC and HPVC.

**Figure 8 materials-14-00985-f008:**
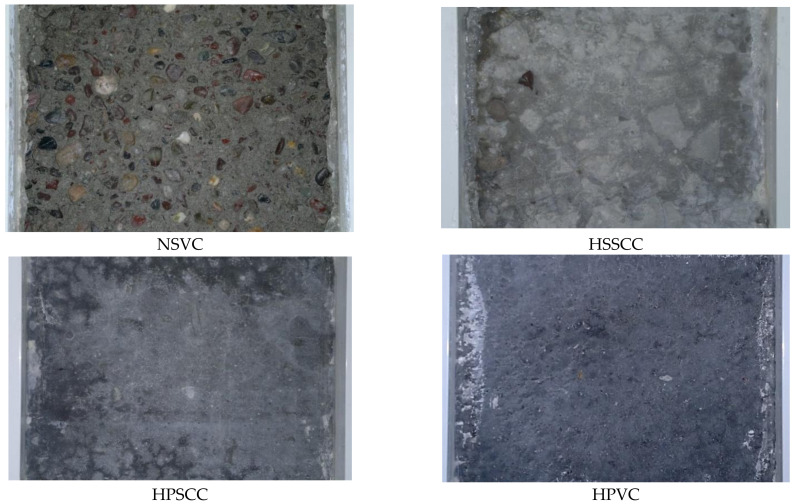
Scaling resistance test of concrete surface exposed to de-icing salts after 50 cycles.

**Figure 9 materials-14-00985-f009:**
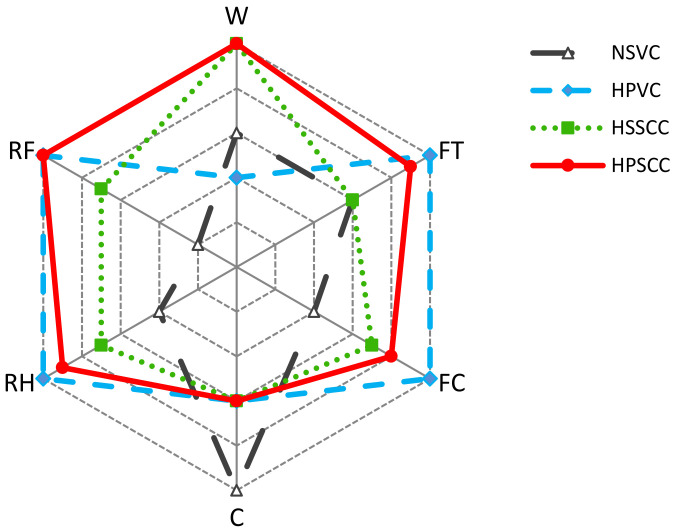
Hexagonal model for multi-parameter comparison of different concrete classes (W—workability measures; FT—tensile strength; FC—compressive strength; C—economic considerations; RH—residual strength after heating to +700 °C; RF—residual strength after 50 cycles of freezing and thawing).

**Table 1 materials-14-00985-t001:** Chemical compositions and physical properties of cement, micro-silica, and fly ash.

Characteristics and Main Oxides	Cement	ASTM C150	Micro-Silica	ASTM C1240	Fly Ash	ASTMC 618
CaO (%)	63.12		0.34		1.43	-
SiO_2_ (%)	23.84		93.11	≥85.0	57.32	[∑ (SiO_2_ + Al_2_O_3_ + Fe_2_O_3_) = 88.9 > 70]
Al_2_O_3_ (%)	4.32		0.62	-	19.88	
Fe_2_O_3_ (%)	3.36		1.28	-	11.67	
MgO (%)	1.38	≤6.0	1.04	-	1.36	-
SO_3_ (%)	1.89	≤3.0	0.34	-	0.79	≤5.0
Na_2_O (%)	-		0.28	-	-	
H_2_O (%)	-		1.08	≤3.0	0.24	≤3.0
Insoluble residue (%)	0.74	≤1.5	-		-	
LOI (%)	1.63	≤3.0	0.83	≤6.0	2.28	≤6.0
Initial setting time (min)	140	≥45	-	-	-	-
Final setting time (min)	245	≤375	-	-	-	-
Compressive strength in 3 days (MPa)	34.1	≥12.0	-	-	-	-
Compressive strength in 7 days (MPa)	42.7	≥19.0	-	-	-	-
Specific gravity	3.15		2.64	-	2.32	-
Fineness (m^2^/kg)	316.2	≥160	21,700	≥15,000	-	-

**Table 2 materials-14-00985-t002:** Grading of coarse and fine aggregates.

Material/Sieve Size (mm)	Coarse Aggregate (%)	ASTM C33-G7 Limits (%)	Fine Aggregate (%)	ASTM C33 Limits (%)
19	100	100		
12.5	94	90–100		
9.5	58	40–70	100	100
4.75	1	0–15	98	95–100
2.36	0	0–5	84	80–100
1.18			64	50–85
0.6			38	25–60
0.3			16	5–30
0.15			4	0–10
0.075			0	0–3
Fineness Modulus			3	2.3–3.1

**Table 3 materials-14-00985-t003:** Mix proportions and compositions for the concrete mixes.

Mix	Cement	Concrete Composition						Concrete Granular Structure				Variation in the Comparison Parameters			
	kg/m^3^	C	S	G	MS	FA	SP	S/G	B/A	W/B	Ad/C	Workability	Strength	Cost	Durability
NSVC (Reference)	316	316	848	1137	-	-	3.16	0.782	0.16	0.60	0.00	VC	Low	Low	Low
HPSCC	433	433	909	1039	35	-	4.33	0.875	0.24	0.35	0.08	SCC	High	Normal	High
HSSCC	396	396	831	950	-	158	3.96	0.875	0.31	0.35	0.40	SCC	High	Normal	Normal
HPVC	457	457	960	1096	55	-	4.57	0.875	0.24	0.23	0.12	VC	Extra-high	High	High

Note: C: Cement; S: Sand; G: Gravel; MS: Micro-silica; FA: Fly ash; SP: Superplasticizer; B/A: Binder-to-aggregate ratio; W/B: Water-to-binder ratio; Ad/C: Admixture-to-cement ratio.

**Table 4 materials-14-00985-t004:** Slump test result of normal strength-vibrated concrete (NSVC) and high-performance highly-viscous concrete (HPVC) mixes, a flow test result of high-performance self-consolidating concrete (HPSCC) and high-strength self-compacted concrete (HSSCC), and restricted flow test result for HPSCC and HSSCC.

Slump test results	Mix		NSVC	HPVC
	Slump	(mm)	190	155
	Stabilization time	(s)	3	8
	Average base diameter	(mm)	405	345
Slump flow test results	Mix		HPSCC	HSSCC
	Segregation Index	(SI)	0	0
	Average flow diameter	(mm)	810	750
	T500	(s)	2.6	3.8
Restricted flow test results	Mix		HPSCC	HSSCC
	Flow diameter	(mm)	760	740
	T500	(s)	3.6	4.1
	ΔH (inside & outside)	Avg. (mm)	3.7	4.5
		Accepted limit (mm)	0–10	0–10

**Table 5 materials-14-00985-t005:** Density of 100 mm cubes and tile shaped specimens, and absorption test results. Abbreviations: High-performance self-consolidating concrete (HPSCC), high-strength self-compacted concrete (HSSCC), normal strength-vibrated concrete (NSVC), and high-performance highly-viscous concrete (HPVC).

Mix	Density of Cubic Specimens (kg/m^3^)					Density of Tile Shaped Specimens at 28 Days (kg/m^3^)		Tile Specimens Difference with the Cubes (%)	Absorption (%), at 28 Days
	28 Days		180 Days		Wt. Loss (%)				
	Avg.	SD	Avg.	SD		Avg.	SD		
HPSCC	2475	13.5	2437	10.2	1.52	2507	8.4	+1.29	0.97
HSSCC	2393	11.9	2381	10.6	0.50	2417	8.0	+1.00	2.09
NSVC	2417	19.3	2357	17.3	2.48	2459	15.6	+1.74	2.32
HPVC	2534	16.9	2483	15.8	2.00	2526	12.5	−0.32	0.57

**Table 6 materials-14-00985-t006:** Compressive strength of 100 mm cubes at 7 ages, and Ø100 mm cylinders.

Mix	Compressive Strength (MPa) of 100 mm Cubic Specimens (fcu) in (t) Days (Gained Strength in Percent Relative to 90 Days’ Strength)							Compressive Strength (MPa) of Ø100 mm Cylinders (fcy) at 90 Days	fcy/fcu at 90 Days
	1 Day	3 Days	7 Days	28 Days	56 Days	90 Days	180 Days		
HPSCC	20.9 (21)	38.6 (39)	58.1 (58)	79.4 (79)	94.4 (94)	100.2 (100)	104.4 (104)	90.3	0.901
HSSCC	8.4 (10)	31.3 (38)	46.0 (55)	59.9 (72)	74.8 (90)	82.9 (100)	86.2 (104)	71.3	0.860
NSVC	4.5 (08)	11.8 (22)	17.5 (33)	34.6 (65)	46.2 (86)	53.6 (100)	55.3 (103)	42.6	0.794
HPVC	30.2 (25)	48.9 (40)	67.8 (56)	103.5 (85)	116.2 (95)	121.9 (100)	126.7 (104)	117.5	0.964

**Table 7 materials-14-00985-t007:** Splitting tensile strength of Ø100 mm specimens, Flexural strength of 75 mm × 75 mm × 350 mm concrete prisms, and Modulus of Elasticity of Ø150 mm specimens.

Mix	Compressive Strength	Splitting Tensile Strength			Flexural Strength				Modulus of Elasticity			
	*f*_cy_ (MPa)	*f*_t_ (MPa)	*f*_ct_/*f*_cy_ (%)	*f*_t_/√*f*_cy_	*f*_r_ (MPa)	*f*_t_ (MPa)	*f*_r_/*f*_t_	*f*_r_/√*f*_cy_	SP1 (MPa)	SP2 (MPa)	E (MPa)	E/√*f*’_c_
HPSCC	90.3	6.21	6.88	0.654	7.15	6.21	1.151	0.752	43.24	42.10	42.7	4.49
HSSCC	71.3	4.60	6.45	0.545	4.82	4.60	1.047	0.571	32.91	32.65	32.8	3.88
NSVC	42.6	4.05	9.51	0.621	4.45	4.05	1.098	0.682	27.94	27.78	27.9	4.27
HPVC	117.5	7.19	6.12	0.663	8.41	7.19	1.169	0.776	45.31	46.00	45.7	4.22

**Table 8 materials-14-00985-t008:** Resistance of specimens exposed to high temperature, and specimens exposed to direct fire.

Mix	The Resistance of Specimens Exposed to High Temperature							The Resistance of Specimens Exposed to Direct Fire			
	Weight (g)			Compressive Strength (MPa)				Compressive Strength (MPa)			
	Dry in 60 °C	Heated to 700 °C	Weight Loss (%)	Control Cubes	Heated Cubes	Residual Strength (%)	Strength loss (%) by Heating (700 °C)	Control Cubes	Cubes Exposed to Fire (500 °C)	Residual Strength (%)	Strength Loss (%)
HPSCC	2462	2317	5.89	93.2	44.9	48.2	51.8	93.2	73.2	78.5	21.5
HSSCC	2401	2238	6.79	68.2	25.4	37.3	62.7	-	-	-	-
NSVC	2403	2222	7.53	44.0	9.4	21.4	78.6	44.0	29.0	65.8	34.2
HPVC	2516	2379	5.45	105.4	53.9	51.1	48.9	-	-	-	-

**Table 9 materials-14-00985-t009:** Loss in mass and compressive strength for 100 mm concrete cubes due to freeze-thaw cycles, Mass losses of concrete specimens the scaling test, and Wearing of the concrete surface due to freeze-thaw cycles.

Mix			HPSCC	HSSCC	NSVC	HPVC
Loss in mass and compressive strength for 100 mm concrete cubes due to freeze-thaw cycles	Mass loss (%) after	10 cycles	0	5.19	5.43	0
		25 cycles	0.02	5.67	22.94	0
		50 cycles	0.02	6.13	82.64	0.02
	Compressive strength (MPa)	Control cubes	109.8	90	56.1	130.2
		After 50 cycles	99.1	56.9	8.1	120.8
		Residual strength (%)	90.3	63.2	14.4	92.8
Mass losses of concrete specimens at the scaling test, and wearing of the concrete surface due to freeze-thaw cycles	Initial mass (kg)	0 cycles	12.591	12.048	11.621	13.026
	Mass loss (%) after	10 cycles	0	0.62	1.87	0
		25 cycles	0.01	0.87	6.83	0
		50 cycles	0.02	1.14	15.81	0.01
	50 cycles weight loss	kg/m^3^	0.387	27.18	372.6	0.286
		kg/m^2^	0.0194	1.359	18.631	0.014
	Scaling depth of exposed surface (mm)		0.010 ≈ 0	0.570 ≈ 1	7.905 ≈ 8	0.005 ≈ 0

**Table 10 materials-14-00985-t010:** Rating of concrete specimens in scaling test ASTM C672.

Mix	Rating/No. of Cycles for the Tested Specimens						Ranking-Surface Conditions According to ASTM C672
	0	5	10	15	25	50	
NSVC	0	1	2	3	4	5	(0) No scaling; (1) Very slight scaling (3 mm depth, max, no coarse aggregate visible); (2) Slight to moderate scaling; (3) Moderate scaling (some coarse aggregate visible); (4) Moderate to severe scaling; (5) Severe scaling (coarse aggregate visible over the entire surface)
HPVC	0	0	0	0	0	0	
HSSCC	0	1	2	2	2	3	
HPSCC	0	0	0	0	0	0	

**Table 11 materials-14-00985-t011:** Comparison scales and rating for the four concrete mixes.

Mix	Strength, 90 Days		Durability, 90 Days		Workability, (mm)	Cost ($)	MPAS		
	(MPa)		(%)						
	Compressive Strength	Tensile Strength	Oven Residual Strength	Residual Strength in F/T Cycles	Slump or Slump Flow	Cost-Effectiveness	Average Out of 10	Hexagon Area	Relative Area
HPSCC	100.2	6.21	48.2	90.3	810	135	-	-	-
S.N.	8	9	9	10	10	6	8.7	196.6	1
HSSCC	82.9	4.6	37.3	63.2	750	125	-	-	-
S.N.	7	6	7	7	10	6	6.1	132.1	0.67
NSVC	53.6	4.05	21.4	14.4	190	80	-	-	-
S.N.	4	6	4	2	6	10	5.3	69.3	0.35
HPVC	121.9	7.19	51.1	92.8	155	140	-	-	-
S.N.	10	10	10	10	4	6	8.3	173.2	0.88

S.N. standardized number.

## Data Availability

No new data were created or analyzed in this study. Data sharing is not applicable to this article.

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
