# Peer review of "Assessment of High Performance Self-Consolidating Concrete through an Experimental and Analytical Multi-Parameter Approach"

_materials, 2021, doi:10.3390/ma14040985_

Round 1
Reviewer 1 Report
This paper reports experimental tests of the fresh and hardened properties of self-consolidating concrete. Although a lot of test data are reported, this paper does not have sufficient technical contribution. I did not see new knowledge that can be learned from this paper. In addition, the quality of presentation is low. A lot of essential details of the experiments are missing, and there is lack of in-depth analysis or discussions of the test data. The quality of figures and tables also needs significantly improvement. Therefore, I do not recommend this paper for publication.
Author Response
Dear Reviewer
Enclosed please see revised version of the manuscript and detailed answers to your valuable comments.

Reviewer 2 Report
In this paper, the performance of self-consolidating concrete is analyzed by various methods and compared with different concrete. Although the topic is relevant, it was deemed to need careful revision. The suggestions are as follows:
Point 1: The literature review should be more consistent with the research theme and confirm the existing research basis
Point 2: I Self-consolidating concrete has been analyzed in many articles.t is necessary to carefully compare the previous articles to refine the innovative points of the article.
Point: A new paragraph needs to be added at the end of the introduction to illustrate the structure of the article.
Author Response

(The authors gave the same response as above.)

Reviewer 3 Report
General comments:
The research presents the evaluation of high-performance self-consolidating concrete.
The article subject is within the scope of the journal. The methodology comprehends a series of physical experiments and a cost estimation.
The experiments are described in detail but the cost estimation is not explained; it only appears in the last table.
The research should give a more important role to this optimization of competitive objectives to provide a wider approach.
The literature review is right but could be improved.
The methodology is focused on the physical experiments and does not refer on how the economic feasibility is considered.
The description of the results is time-consuming for the reader and should be synthesized.
The paper needs proofreading. The recommendation of this reviewer is to make major corrections for a next review.
Specific comments
Line 76: the authors should clearly state the aim of the research here or in the other section but only once.
Line 89: there is no reference to NSC
Lines 111-112: reword sentence Lines 128-129: it is not clear how Table 3 shows the optimized parameters (workability, strength, cost, and durability)
Lines 140-142: reword the first sentence. Lines 152: explain what T500 is
Lines 187-188: the oven´s capacity is not the temperature it reaches. Reword.
Lines 200-201: cite the source for this statement. Lines 218-220: reword Lines 224-226: reword Lines 228-236: reword
Lines 243-247: reword
Lines 307-308: reword
Lines 377-379: reword
Lines 392-398: reword
Lines 411-414: reword
Lines 421-424: reword
Lines 471-472: reword
Line 484-485: reword, define QC
Line 501: m3 (superindex)
Lines 509-542: results should be in the former section, no future lines of research are mentioned, conclusions should be improved.
Author Response

(The authors gave the same response as above.)

Reviewer 4 Report
I suggest this article be rejected. Its very long, confusing, and I am not too sure that its very useful. Please see below for my comments:
- Language is confusing and the paper is hard to understand. A review from a native speaker may be needed.
- L73: Not sure what the hypothesis was.
- Introduction and literature review is pretty generic. Much of this material is out there. I am not quite sure what is new here?
- A lot of the literature review and the references appear to be too general or not really relevant here.
- Table 3: Multiple parameters are changing here and I am quite sure what you intend to compare?
- Fig. 1 to 4, 7, 8 are not needed. Nothing novel and poor quality.
- Referencing needs to be improved.
- L272: But fly ash reacts in concrete and that is the whole point of using it. Incorrect statement.
- Does a 1% change in density really matter?
- You cannot simply compare strength of SCC vs. normal concrete, the type of admixtures used will have a huge impact.
- Table 11: This seems very qualitative. Selection considerations are largely driven by local factors and also by location-specific requirements. So simply stating that HPSCC is the best mixture is missing a lot of nuance.
- I fear the conclusions are very specific and not generalizable.
Author Response

(The authors gave the same response as above.)

Round 2
Reviewer 1 Report
The quality of the revised version is improved but further improvement is needed. Following are some detailed comments:
- The introduction section needs to be improved to clearly state the research problem of this paper. You mentioned UHPC and compared HPC against UHPC. The statement is misleading because you only briefly mentioned the cost, which is inappropriate. If you want to compare two materials, the performance must be considered. Please revise the related statements and update the old references using the following two new references of self-consolidating UHPC: Khayat, K.H., Meng, W., Vallurupalli, K. and Teng, L., 2019. Rheological properties of ultra-high-performance concrete—An overview. Cement and Concrete Research, 124, p.105828. Meng, W. and Khayat, K.H., 2018. Effect of graphite nanoplatelets and carbon nanofibers on rheology, hydration, shrinkage, mechanical properties, and microstructure of UHPC. Cement and Concrete Research, 105, pp.64-71.
- The section 2 (Review of the literature) can be merged into the section 1 (Introduction). The paragraph that introduces the structure of the paper can be removed: "This paper includes 6 sections, which ..."
- In Table 3, you listed the ratios, which are not clear. Please replace the ratios using mass per cubic meter for all the components. Also, you must introduce the meanings of the symbols used to stand for the different material components. In addition to SP, did you use any VMA? Did you test the rheological properties? Did you observe any segregation?
- For each figure, a caption must be used before you introduce the meaning of each image. See Fig. 1. In addition, the quality of your photos is low. Most photos are small and blur.
- The fire resistance is not a durability property. You should list it as a separate property. You can call it fire resistance. In the related results and discussion in section 4, please discuss the mechanisms of the mechanical properties. Refer to: Li, et al, 2017. Thermal and mechanical properties of high-performance fiber-reinforced cementitious composites after exposure to high temperatures. Construction and Building Materials, 157, pp.829-838.
- In the radar plot in Fig. 9, the exact numbers of each axis should be included. How did you evaluate the economic considerations? Did you calculate the costs? If yes, please list the inventory that you used in the calculation.
- The language should be polished. There are many ambiguous expressions.
Author Response
The authors highly appreciate the reviewer's comments, which definitely led to improvement in our manuscript.
To find the authors' response to the reviewer's comments, please see the attached file.

Reviewer 3 Report
This reviewer acknowledges the effort that the authors made to improve the paper´s quality however it still needs proof-reading.
There are some corrections to be done:
Line 139: Table 3 can explain that 3 mixes were of the same proportions among (not between) cement, sand and gravel, while NSVC is a reference mix.
Line 187: prisms (not prims)
Line 224: “The scaling test covers determination of the resistance to scaling of a horizontal concrete surface exposed to 50 freeze/thaw cycles in presence of de-icing chemicals and it is intending to evaluate concrete surface resistance qualitatively by visual examination, as in ASTM C672 [55]. Reword.
Line 237: calculated (not found)
Line 238: “The data for economic assessment considerably vary between regions”. (data is plural)
Line 238: This is because local conditions highly affect the cost of labour and market costs for recovered materials, as well as transportation scenario. Reword.
Line 247: “it was continued in spreading in a slower rate”. Reword
Line 335: “Combined effects of the specimens’ shape and casting direction during tests, were the two factors that controlling (fcy/fcu) ratio. Reword
Line 339: increase (not increases)
Line 382: since the composition of the mixtures plays a (not have) great role
Line 405: “The limit of which transfers the heating from a useful factor to a destructive
factor, could depends on many factors, however, in general the temperature higher than 400℃ regarded as disadvantageous”. Reword
Author Response

(The authors gave the same response as above.)

Reviewer 4 Report
- Language still needs quite some work.
- Novelty is still not clear.
- Font sizes appear to be inconsistent.
- Not sure how Fig. 1-4, 6-8 help.
- Quality of the figures is poor.
Author Response

(The authors gave the same response as above.)
